# Recent collisional history of (65803) Didymos

Adriano Campo Bagatin [1,2] ✉, Aldo Dell'Oro [3], Laura M. Parro [1,2], Paula G. Benavidez [1,2,4], Seth Jacobson [5], Alice Lucchetti [6], Francesco Marzari[7], Patrick Michel [8], Maurizio Pajola [6] & Jean-Baptiste Vincent [9]

The Double Asteroid Redirection Test (DART, NASA) spacecraft revealed that the primary of the (65803) Didymos near-Earth asteroid (NEA) binary system is not exactly the expected spinning top shape observed for other km-size asteroids. Ground based radar observations predicted that such shape was compatible with the uncertainty along the direction of the asteroid spin axis. Indeed, Didymos shows crater and landslide features, and evidence for boulder motion at low equatorial latitudes. Altogether, the primary seems to have undergone sudden structural failure in its recent history, which may even result in the formation of the secondary. The high eccentricity of Didymos sets its aphelion distance inside the inner main belt, where it spends more than 1/3 of its orbital period and it may undergo many more collisions than in the NEA region. In this work, we investigate the collisional environment of this asteroid and estimate the probability of collision with multi-size potential impactors. We analyze the possibility that such impacts produced the surface features observed on Didymos by comparing collisional intervals with estimated times for surface destabilization by the Yarkovsky-O'Keefe-Radzievskii-Paddack (YORP) effect. We find that collisional effects dominate over potential local or global deformation due to YORP spin up.

On the 26th of September, 2022, the Didymos Reconnaissance and Asteroid Camera for OpNav (DRACO) scientific camera on board the Double Asteroid Redirection Test (DART, NASA) spacecraft took images of the primary of the near-Earth asteroid binary (65803) Didymos, a few minutes before the spacecraft intentionally crashed into Dimorphos, the secondary body of the system, in the first asteroid impact deflection experiment ever performed[1,2].

Images of Didymos show a not completely axisymmetric body. The LICIACube (ASI) cubesat top-down images show a squarish equatorial region. Didymos has an equatorial smooth (at available resolution) bulge, with many large boulders on its visible side. Images also reveal several crater-like structures located away from the equatorial region, evidence of landslides, and the presence of many linear North-South and South-North tracks in both northern

[1]Instituto de Física Aplicada a las Ciencias y las Tecnologías, Universidad de Alicante, Ctra. de San Vicente del Raspeig, s/n, Sant Vicent del Raspeig 03690 Alicante, Spain. [2]Departamento de Física, Ingeniería de Sistemas y Teoría de la Señal, Universidad de Alicante, Ctra. de San Vicente del Raspeig, s/n, San Vicente del Raspeig 03690 Alicante, Spain. [3]Osservatorio Astrofisico di Arcetri, INAF, Largo Enrico Fermi, 5, Firenze I-50125, Italy. [4]European Space Astronomy Centre, European Space Agency, Camino bajo del Castillo S/N, Urbanización Villafranca del Castillo, Villanueva de la Cañada, Madrid 28692, Spain. [5]Department of Earth and Environmental Sciences, Michigan State University, 288 Farm Ln. Natural Science Bldg, East Lansing, MI 48824, USA. [6]Astronomical Observatory of Padova, INAF, Vic. Osservatorio 5, Padova 35122, Italy. [7]Department of Physics and Astronomy 'Galileo Galilei', Università degli Studi di Padova, via Marzolo 8, Padova 35131, Italy. [8]Laboratorie Lagrange, Observatoire de la Côte d'Azur - CNRS, Cedex 4, Nice 03604, France. [9]DLR Institute of Planetary Research, Rutherfordstrasse 2, Berlin 12489, Germany. ✉e-mail: acb@ua.es

and southern equatorial regions, with and without boulders at their end.

A shape model of Didymos was available from ground-based optical and radar observations before impact. The top-shape had large error bars along the z-axis. They predicted that Didymos shape was compatible with a top shape, given the large error bars along the spin axis, similar to other asteroids targeted by space missions in the last decade, like Ryugu (visited by the JAXA spacecraft Hayabusa 2), Bennu (visited by the NASA spacecraft OSIRIS-REx), as well as primaries of other binary systems observed by radar. Didymos looks rather like a degraded top–shape, with a very fast spin rate, $T = 2.2600$ h, at the disruption spin barrier limit. Initial estimations of the body equatorial extents[3] implied a non-equilibrium state[4] (in the sense of equatorial instability). However, recent estimation of its mass, volume and axes of the equivalent volume ellipsoid[5] place the object in a region of the mass–volume parameter space compatible with both equilibrium and equatorial instability.

The Didymos system orbits the Sun with an eccentric orbit ($e = 0.38375$) driving it as far as $Q = 2.2755$ au (aphelion distance) from the Sun, well inside the inner main asteroid belt. This makes the system spend 1/3 of its orbital period inside that region.

At first glance, the YORP (Radziewsky-O'Keefe-Rubincam-Paddack) non–gravitational spin acceleration[6,7] may be indicated as the culprit for the degradation features on Didymos[8]. The YORP effect depends on a number of unknowns; still, it has been already measured on several asteroids over a decade time scale, indicating generic spin up (e.g. ref. [9]), except for the case of Ryugu, for which surface evolution indicates spin down[10]. It is important to be aware that–on longer time scales (>10–100 ky)–this effect cannot be just assumed as constant. In fact, as pointed out by refs. [11,12], small surface modifications may trigger changes in the torque produced by the solar radiation pressure forces on the asteroid, even leading to switch from positive to negative angular acceleration.

We envision two possible origins for the degraded surface features on Didymos. On the one hand, given the high spin rate of the asteroid, one possibility is that shear stress is built up by YORP acceleration at subsurface level until it overcomes friction yield, eventually producing landslides. On the other hand, if the impact frequency of small impactors is high enough, then relatively low-energy collisions may be more efficient in triggering landslides, surface deformation and/or cratering. Collisions in the asteroid belt have already been indicated as the main source of asteroid spin up[13,14], so it would not be surprising that they are also responsible for surface degradation.

In this work, we analyze the impact probability of the system considering: a) the real population of objects crossing its orbit (the Didymos Crossing Orbit, DCO, population) down to completitude, and b) the size distribution of smaller objects, which are those that may impact Didymos and Dimorphos. Such population was derived matching available modeled size frequency distributions for the asteroid belt[15] and the DCO populations, by multiplying the first by a suitable scaling factor. As a result, average impact time intervals are derived, as a function of the impactor energy and size. We find that the observed surface degradation features observed on Didymos are most probably due to impacts rather than to the effect of YORP-induced spin up.

## Results
In this study, we estimate the time scale of slow buildup of shear stress due to assumed constant positive spin up, and compare it with the estimated average impact times due to the transit of Didymos inside the inner asteroid belt. To do so, we calculate impact probabilities on Didymos and Dimorphos and introduce the possibility that the current Didymos shape and surface morphology is due to impact events rather than to the constant gentle action of YORP[8].

## Building up shear stress by YORP
The YORP effect is a non-gravitational torque based on asymmetric solar radiation re-emission, that may spin up or down small asteroids[6,7]. Were the YORP effect responsible for producing the observed features on Didymos, that might happen through slow buildup of shear stress due to increasing angular velocity. The stress should then be released suddenly at some time, when the sub-surface asteroid friction yield is overcome. The YORP acceleration on Didymos is unknown, so any parameter choice for its modeling is arbitrary. An order of magnitude estimation may be tried though. Instead of taking the smallest or larger values of YORP acceleration measured on asteroids, the measured YORP acceleration on asteroid Ryugu may be used. The reason behind assuming Ryugu as a proxy to Didymos is that their moments of inertia with respect to their spin axes can be related. Ryugu is actually 20% larger than Didymos and the two asteroids have different composition: Ryugu is a C-type[16], and Didymos is an S-type[17]. From its equatorial size and mass, Ryugu's moment of inertia is estimated as $I_R \approx 1.7 \times 10^{16}$ kg m$^2$. The corresponding Didymos moment of inertia ($I_D$) is $I_D \approx 0.7\,I_R$. Ryugu has been estimated to have an angular (negative) acceleration of $\alpha = d\omega/dt = -(0.42{-}6.3) \times 10^{-6}$ deg/day$^2$ [10]. Didymos may be estimated to have a larger angular acceleration due to its smaller moment of inertia, and may be assumed to be $(0.5{-}10) \times 10^{-6}$ deg/day$^2$, for our purpose.

During a given time interval $\Delta t$, the spin rate change, $\Delta\omega = \alpha\,t$, increases the shear stress per unit mass, $\Delta\sigma/m$, normal to the spin axis, in the non-inertial reference system of the rotating asteroid, by a given amount: $\Delta\sigma/m = 2\omega\,\Delta\omega\,r/S$ (S is any surface area normal to the direction of the spin axis), neglecting terms in $(\Delta\omega)^2$ ($r$ is the distance from some near-surface point to the spin axis). Let us make reference to some point close to the surface of the asteroid at mid latitudes, for which no net shear stress is acting at the beginning of the spin up process starting at some angular velocity $\omega_0$. It turns out that the effect of spin up on the shear stress per unit mass over 1 My is, at most, $\Delta\sigma/m \approx 1{-}2 \times 10^{-4}$ Pa/kg, which is comparable to the friction forces per unit area and mass at those asteroid latitudes. In terms of cohesion, the built up shear stress requires a cohesion of 0.25–0.50 Pa at 1 m depth, which may be provided by friction. This means that friction forces may be able to withstand the accumulation of shear stress over about 1 Myr, at some depth, before it can be suddenly released and cause mass motion in form of landslides. This calculation does not even take into account other potential sources of stiffness, like interlocking and inter-particle cohesion (if any), and does not consider the potential damping effect of collisions on YORP[13], nor any effect due to the binary YORP[18].

In this situation, any perturbation on the surface within such time range, like low energy impacts on the surface, shall release local shear stress due to spin up, resetting stress buildup and avoiding large scale deformation that would –instead– take place on a collision-free Didymos.

## The collisional environment of the Didymos system
Along its eccentric orbit, Didymos crosses three different asteroid populations: the Near-Earth Asteroids (NEA), the poorly populated Hungaria asteroid region and the inner belt, where it spends 1/3 of its orbital period. The contribution to collisions is expected to be dominated by the inner belt asteroids as they outnumber both the NEA and the Hungaria populations.

In order to compute the relevant statistical parameters of the effective collisional environment of the Didymos/Dimorphos system– mainly impact frequency and distribution of the impact velocity–we restricted our analysis to the population of impactors whose orbits can cross the orbit of Didymos, whatever region they come from. We name such population as Didymos Crossing Objects (DCO). The number of impacts per unit time suffered by Didymos or Dimorphos from DCO

impactors with diameter larger than $D$ is given by:

$$f = \frac{1}{4}(D_T + D)^2 \langle P_i \rangle N(>D) \qquad (1)$$

where $D_T$ is the diameter of the target (Didymos or Dimorphos), $N(>D)$ is the number of DCOs with diameter larger than $D$, and $\langle P_i \rangle$ is the mean intrinsic probability of collision between the target and DCO impactors. The latter parameter has been introduced by ref. 19, and it is related to the target orbit and the distribution of impactor orbits. As shown in detail in the "Methods" section, we estimated that $\langle P_i \rangle = (0.9-1.2) \times 10^{-17}$ km$^{-2}$ yr$^{-1}$.

For what concerns the impactors distribution, $N(>D)$, we assume that it is the same as that of the Main Asteroid Belt (MAB), $N_{MB}(>D)$, suitably rescaled to take into account that DCOs include only a fraction of MABs. In other terms: $N(>D) = k\,N_{MB}(>D)$, where $k$ is a constant factor. As $N_{MB}(>D)$, we used the debiased size distributions computed by ref. 15. They propose different possible distributions compatible with MAB observables, for the size range below 1 km. We used the two extreme cases (n. 1 and n. 8 in ref. 15, according to different values of the impact-specific dispersion energy in their collisional model), corresponding to the lower and upper limits for $N(<D)$. We rescaled the two MAB debiased distributions in order to match the observed DCO size distribution in the range between 1 and 10 km. Including the uncertainty on the albedo values of DCOs, we estimated that the value of factor $k$ should be between $0.26 \pm 0.01$ and $0.31 \pm 0.01$. The DCO populations are represented in Fig. 1.

Finally, the distribution of impact velocities between the Didymos system and the DCO impactors population is closely related to the computation of $\langle P_i \rangle$, depending essentially on the target orbit and the distribution of the projectile orbits (see the "Methods" section for details). The computed impact velocity distribution is shown in Fig. 2, and the mean impact speed $\langle U \rangle$ is about 7.5 km/s, which is higher with respect to average MAB relative speeds (Fig. 2).

### Impact time intervals for Didymos and Dimorphos

Once the characteristics of the collisional environment of the Didymos system are established, we can compute the impact frequency, $f$, as function of the impactor size $D$. It is also interesting to study the

characteristic time, $\tau = 1/f$, of the impact with projectiles which kinetic energy is larger than a given value $E$. Assuming that the typical relative speed $U$ is the mean impact velocity determined above, a collision with energy larger than $E$ is due to an impactor whose diameter is larger than:

$$D = \left( \frac{12}{\pi} \frac{E}{\rho \langle U \rangle^2} \right)^{1/3} \qquad (2)$$

where $\rho$ is the impactor mass density, assumed as an average value of 2.5 g/cm$^3$. The result of this exercise is shown in Fig. 3, where the characteristic impact time intervals on Didymos and Dimorphos are plotted versus the impact energy in terms of the impact energy of the DART experiment ($E_{DART} = 1.094 \times 10^{10}$ J$^3$), and versus the impactor size. The equivalent diameter of targets is assumed equal to 730 m and 150 m for Didymos and Dimorphos, respectively[5] (and DART DRA 5.20).

## Discussion

DART energy impacts occur on average on Didymos every 73–84 kyr, depending on the choice of the scaling factor. Assuming that Didymos has spent already half of a median NEA lifetime (8–10 Myr)[20] in its current orbit, it should have undergone tens of DART-like impacts in that period of time. The effect of such impacts on the Didymos surface is unknown but it may have triggered landslides due to its fast spin rate (2.26 h), modifying its appearance and surface morphology, and making it hard to extrapolate current surface features to surface ages. In fact, dating the surface age after some major resurfacing event (e.g. the latest large impact) may be misleading because later, low-energy impacts may produce surface modifications comparable to those caused by the impact itself.

Didymos shows at least one big feature ranging 268 m in size, interpreted as an impact crater[21], which is 32% of the asteroid diameter. Largest craters on asteroids are almost never larger than 40% the asteroid diameter, therefore the effect of such an impact was likely to produce large scale effects on the asteroid morphplogy. So, for Didymos this impact was rather big and certainly had large scale effects. Assuming standard scaling laws for cratering in the strength

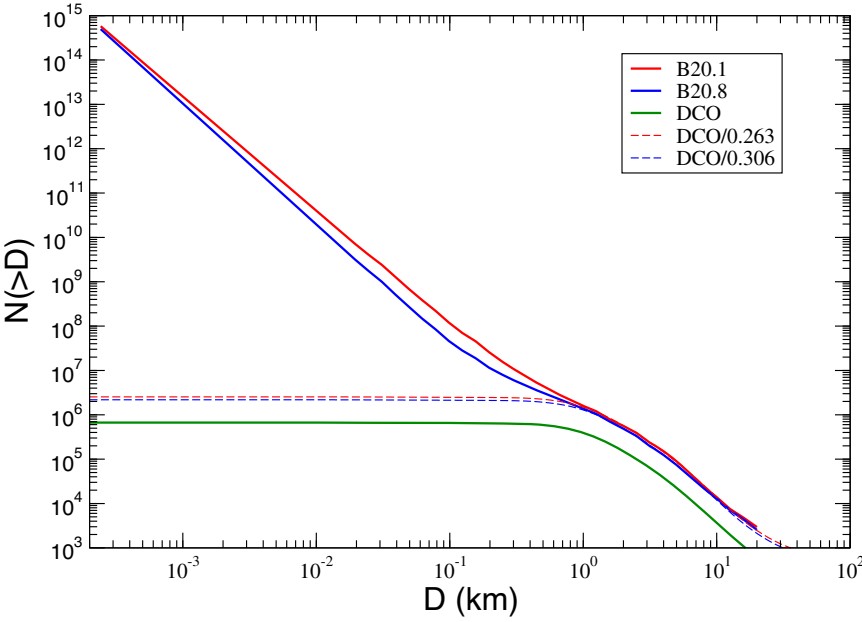

**Fig. 1 | DCO (Didymos Crossing Objects) size-frequency distributions.** Biased distribution $N'(>D)$ of DCO (green) and[15] de-biased distributions of MAB n. 1 (red) and n. 8 (blue). Dashed lines are the re-scaled distributions $N'(>D)/0.263$ (red) and $N'(>D)/0.306$ (blue). Source Data are provided as a Source Data file.

regime, that would correspond to about a 10 m size impactor. This is a very uncertain estimation of the impactor size though, in fact, on the one hand, the projectile needed to make such a crater may be even smaller on a granular surface in the gravity regime. On the other hand, a larger projectile may have the same effect in an armoring regime, where the impactor hits a similar-size boulder on the surface, dissipating part of its energy before generating the crater[22]. There are ten crater-like features identified with the same level of confidence on Didymos, and a few more uncertain ones.

Our calculation for the average impact rate of 10 m projectiles gives $\tau_{DD} = 75$–130 Myr. Nonetheless, we need to keep in mind that Didymos current orbit is not expected to have lasted more than the median NEA lifetime ($\approx 8$–10 Myr) so far, so $\tau_{DD}$ is clearly an overestimate, and it is not straightforward to extrapolate an impact frequency for Didymos back to when it still was in the main belt. In fact, we do not know what region of the main belt it came from, which may affect its average encounter speed and the population density of potential impactors. One can try to have an idea of such average impact time by simply taking into account the assumed Didymos age as a NEA, and the fact that Didymos currently spends 1/3 of its orbital

period inside the inner belt, where the impact probability is significantly larger than in the NEA region. This results in an estimation of one 10 m projectile impact every 25–45 Myr, and a 95% Poisson probability to have at least one such impact over 75–130 My. None of those time intervals should be taken as estimates of the Didymos surface age. In fact, as previously discussed, surface modification may happen at any time later than the largest impact, due to low-energy impacts triggering local damage and landslides.

Dimorphos size is very close to the minimum values of scaling laws for disruption (e.g., see Fig. 2 in ref. [15]), as reported for coherent, monolithic targets. Its size is in the transition zone between strength and gravity-dominated objects. Dimorphos internal structure is unknown (and shall be measured by the ESA Hera mission[23]), but its appearance and the boulder SFD are compatible with a rubble-pile structure[3,24]. Numerical modeling[25] recently found that the disruption-specific energy for rubble piles the size of Dimorphos is in the 145 to 1140 J/kg range. In terms of the DART energy ($E_D$), it turns out that impacts with at least 65 $E_D$ can disrupt objects the size of Dimorphos. We applied the standard relationship for the size $d$ of any given impactor able to disrupt a given target of size $D$: $d/D = (2Q_D^*/V_{rel}^2)^{1/3}$, assuming the same density for both bodies (e.g. ref. [15]), and we obtained that the critical size of a projectile able to disrupt Dimorphos is at least 2.6 m. Therefore, we find that the average time between two disrupting collisions on Dimorphos is in the range 50–70 Myr, depending on the value of the re-scaling parameter in our DCO extrapolated size distribution. Again, this is based on the current orbit of the system, and this time widely exceeds the median expected NEA lifetime. Using the same argument as in the case of Didymos, we can try a gross estimation of the impact interval on Dimorphos during its stay in the MAB, considering that the system was spending three times as much time inside the asteroid belt than during its current orbit. In this way, we estimate the average minimum lifetime of Dimorphos as 20–30 Myr. Consequently, a 95% Poisson probability of survival for Dimorphos corresponds to an age estimate of 2.5–3.5 Myr. Unfortunately, this does not clarify whether the Didymos system was formed while the parent Didymos was already in the NEA region, or it rather still was in the MAB. Recent modeled surface age was estimated to be 0.3 Myr[21], which is extremely young, and it may indicate that the binary formation took place when the Didymos parent body was already a NEA.

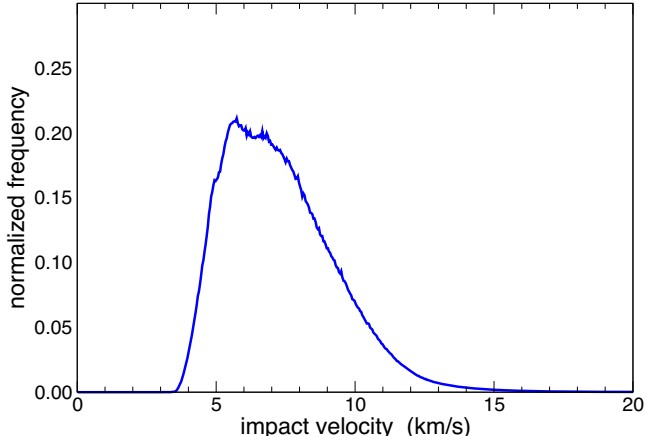

**Fig. 2 | Distribution of the relative velocity of DCO impactors with respect to the Didymos system.** Source data are provided as a Source Data file.

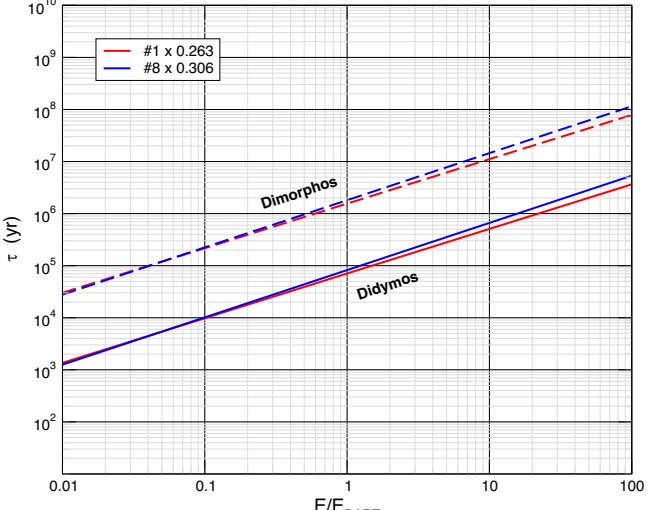

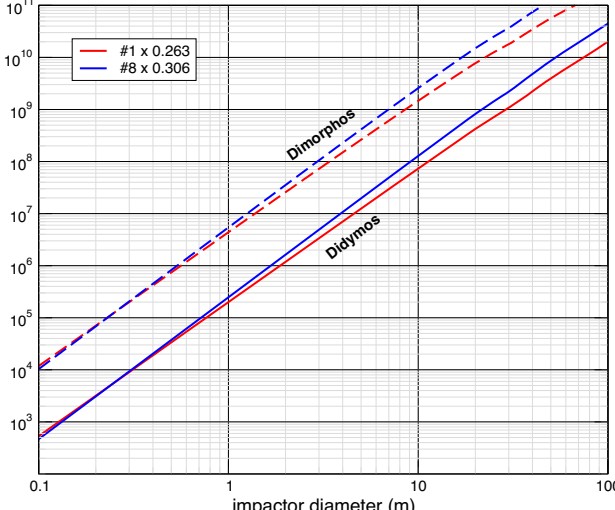

**Fig. 3 | Average impact time intervals.** Characteristic average time interval $\tau$ for impacts on Didymos and Dimorphos with projectiles having relative kinetic energy larger than $E/E_{DART}$ (left), and larger than a given size (right). $E_{DART}$ is the impact kinetic energy of the DART impact. Number "1" and "8" refer to the extreme cases of the debiased Main Belt Asteroids size distribution by ref. [15]; 0.26 and 0.31 are the corresponding values of the rescaling parameter $k$ (see text for details). Source Data are provided as a Source Data file.

As for smaller impacts, like the $E_D$ ones, we find an average interval of 1.55–1.85 Myr. That means that Dimorphos very likely already underwent DART-energy impacts in its lifetime, and for sure it underwent tens of less energetic ones, as can be checked in Fig. 3. This, in turn, may trigger swarms of low-energy ejecta impacts on Didymos, both due to direct high-speed impacts and low speed boulder re-impacts, as recently discussed[26,27]. It will be a matter of future investigation to study the effect of such impacts on Didymos over time. Partial destabilization of the surface may be expected[28], and add to direct impacts on the primary.

In "Building up shear stress by YORP", we showed that it takes at least 1 Myr to build up stress nearby the surface to the level to overcome friction forces. Our collisional probability calculation shows that in that time, Didymos may undergo some 100 low energy impacts (at least $0.1E_D$), more than 10 at least DART-energy impacts (or larger), and possibly one impact at least 10 times more energetic. (See Fig. 3). In conclusion, such impact induced surface perturbations are very likely responsible for releasing stress on the surface produced by YORP spin up, causing themselves the landslides and surface features that have been identified on the Didymos surface. This result clearly points at collisions as the dominant source of surface evolution in the Didymos system.

The main dynamical difference between the Didymos binary system and other small asteroids imaged by spacecraft, like Ryugu and Bennu, is that part of its orbit enters the inner belt, where collisional probability increases dramatically. On the contrary, Ryugu and Bennu have orbits well inside the NEA region with no interaction with the main belt asteroids ($q_{Ryugu} = 0.963$ au, $Q_{Ryugu} = 1.42$ au; $q_{Bennu} = 0.897$ au; $Q_{Bennu} = 1.36$ au). This circumstance, together with the extremely high spin rate of the primary, are the main peculiarities of the Didymos system, which likely led to its current shape, departing from the top shape shown by both Ryugu and Bennu.

Such considerations drive more general questions. Do small NEAs go through a transition phase, when their eccentricity is still high so that they are still subject to collisional interaction with the inner belt asteroids, before coming to a quiet environment entirely inside the NEA region? Do they get their top shape in their latter phase, just under the action of YORP? We don't think we have the answer to these questions at the moment. In this context, the images taken by the Lucy (NASA) spacecraft of the inner main belt binary asteroid Dinkinesh in early November 2023, may add to this discussion, and the Dinkinesh primary seems to be yet another top shape. On top of that, small asteroids (~1 km) may be driven by collisions to spin up to their critical period while in the main belt, without invoking any YORP contribution[14].

It is not the goal of this work to investigate the formation of binaries, though it is indeed an interesting debate. Both YORP and collisions may cooperate in the spin up process, and different paths may lead to different outcomes. For instance, we may speculate that such process may result in the formation of a binary with a top shape primary, or rather in the fission of as much mass as it takes for the remaining body to withstand rotational energy and angular momentum, leading to a single object (or even an asteroid pair[29]). The process leading to any such outcome may be the action of YORP, or even a final impulsive event (a small energy collision, a planetary close fly-by), on a parent asteroid already rotating close to its critical spin rate.

## Methods

### Computation of the intrinsic probability and impact speed distribution

Statistics of impacts is provided in terms of the mean intrinsic probability of collision, the mean impact velocity and the distribution of impact velocity, all strictly related to the target orbit and the orbital distribution of the impactors population. In the present study, we focus on the statistics of impacts between the Didymos system and the

population of Didymos Crossing Objects (DCO). For this case, the computation of the statistical parameters of impacts is based on the following dynamical hypotheses:

- the semimajor axes $a$, eccentricities $e$ and inclinations $I$ of the target and impactor orbits are fixed;
- the arguments of pericenters and longitudes of the nodes change uniformly with time;
- the arguments of pericenters and longitudes of the nodes of different orbits are not cross-correlated;

In dynamical terms, the conditions above are valid under the assumption that the values of forced eccentricities and forced inclinations due to secular perturbations are much smaller than the corresponding proper elements. For Solar System asteroids this assumption is not strictly correct, but it is a good approximation for the purposes of this work. Moreover, in this case, deviations in the probability of collision due to secular effects entail errors in the collisional rates smaller than the errors coming from other sources of uncertainty, mainly from our limited knowledge of the real size distribution of impactors.

The intrinsic probability of collision $P_i(a, e, I)$ between the target and any given projectile with orbital elements $a$, $e$ and $I$ is defined as the mean number of close encounters occurring per unit of time within a distance of 1 km, and it is expressed in units of $km^{-2} yr^{-1}$ [19]. For an impactor which orbit cannot cross the target orbit, $P_i(a, e, I) = 0$.

The mean intrinsic probability of collision between the target and the impactors population is defined as:

$$\langle P_i \rangle = \int P_i(a, e, I)\psi(a, e, I)\, a\, e\, I \tag{3}$$

where $\psi(a, e, I)$ is the (normalized) distribution of the impactors orbital elements.

The parameter $\langle P_i \rangle$ is numerically evaluated as the mean of $P_i(a, e, I)$ computed for a set of $N$ orbits representative of the impactors population, using the algorithm by[30], which was validated exploiting the updated and independent approach developed[31].

Following the definitions above, the mean number of impacts per unit time between the target and projectiles larger than size $D$ is:

$$\frac{dn}{dt}(>D) = \frac{1}{4}\langle P_i \rangle (D_T + D)^2 N(>D) \tag{4}$$

where $D_T$ is the diameter of the target and $N(>D)$ is the cumulative distribution of impactors larger than a given size $D$.

The distribution of the impact velocity, like the intrinsic probability, is stricly related to the distribution of the orbital elements, and it is a side product of the methods described in previous work[30,31] (please, see those papers for details). In particular, the value of the mean impact velocity can be written as:

$$\langle U \rangle = \frac{\int U(a, e, I)P_i(a, e, I)\psi(a, e, I)\, a\, e\, I}{\int P_i(a, e, I)\psi(a, e, I)\, a\, e\, I} \tag{5}$$

where $U(a, e, I)$ is the mean impact velocity between the target and a projectile with orbital elements $a$, $e$ and $I$.

A crucial point in the computation of the intrinsic probability and the impact speed distribution is the potential biases in the distribution of the orbital elements of impactors. In order to evaluate the impact of the observational bias on the computation of collisional parameters, we computed them selecting impactors with absolute magnitude less than a maximum value $H_{max}$, and repeating the computation for different values of $H_{max}$. In Fig. 4 the cumulative distribution of the DCOs is shown, along with the same distributions for the Main Asteroid Belt (MAB) and Near-Earth Asteroid (NEA) region, for comparison. Instead,

in Fig. 5 the computed values of $\langle P_i \rangle$ and the mean impact speed $\langle U \rangle$ are plotted as a function of $H_{max}$. An increasing trend for $\langle P_i \rangle$ with $H_{max}$ is evident, and is clearly due to the fact that including increasingly fainter objects we are selecting preferentially orbits with smaller semimajor axes, which contribute with higher values of $P_i(a, e, I)$ in integral (5), as they are closer to the Didymos system.

The value of $\langle P_i \rangle$ varies from $0.9 \times 10^{-17}$ km$^{-2}$ yr$^{-1}$ for $H_{max} = 13$, to $1.2 \times 10^{-17}$ km$^{-2}$ yr$^{-1}$ for $H_{max} = 20$, which is beyond the limit of completeness of the DCO population. Higher values of $\langle P_i \rangle$ are clearly more and more unreliable.

The value of $\langle U \rangle$ is more stable, settling around 7.5 km/s, because for increasing $H_{max}$, the increase of impact probability, due to the increasing number of internal orbits, is balanced by the lower relative velocities of those orbits with respect to the target. Indeed, higher impact velocities are due to the contribution of orbits with higher eccentricities and inclinations, belonging to the external part of the DCO region. However, the census of such orbits is affected by strong observational biases.

The census of DCOs is probably complete until absolute magnitude 16, beyond which the slope of the distribution decreases. A reasonable hypothesis is that the orbits of DCOs with $H < 16$ are

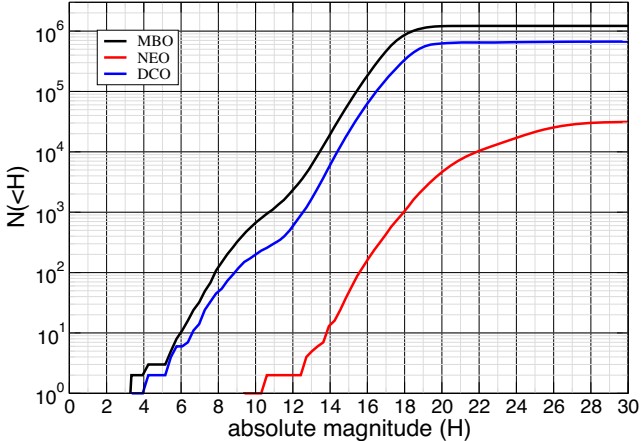

**Fig. 4 | Cumulative distribution of absolute magnitudes (H) of Main Asteroids Belt (MAB), Near Earth Asteroids (NEA) and Didymos Crossing Objects (DCO).** Source Data are provided as a Source Data file.

representative of their global orbital distribution. It is worthwhile pointing out that the calculation of parameters related to the collisional evolution of the Didymos-Dimorphos system take into account the population of DCOs only, excluding bodies with $P_i(a, e, I) = 0$. This applies to both the estimation of the mean intrinsic probability of impact and to the size distribution of impactors (see next section). In conclusion, from Fig. 5 we can state that our best estimation of $\langle P_i \rangle$ is about $10^{-17}$ km$^{-2}$ yr$^{-1}$, with 10% uncertainty.

Let's mention that a different approach was followed in other contexts, also including non crossing orbits in the evaluation of the mean $\langle P_i \rangle$[32]. This explains why the value of $\langle P_i \rangle$ worked out in this work looks significantly higher than typical values of the mean intrinsic probability of collision computed for the whole MBAs. The two approaches are equivalent if the value $\langle P_i \rangle$, in Eq. (4), is calculated for the population of objects corresponding to the size distribution $N(>D)$. The advantage of excluding non crossing orbits is that the computation of the mean intrinsic probability of collision is then independent of any assumption about the orbital distribution of objects that do not interact with the target.

### Model of the impactors size distribution

The population of the DCOs consists, for the large majority, of asteroids belonging to the inner zone of the Main Belt. For this reason, we model the DCO size distribution assuming that it is the same as MAB but suitably scaled in order to take into account that only a fraction of MABs can collide with the Didymos/Dimorphos system. In other terms, if $N(>D)$ is the cumulative size distribution of DCOs and $N_{MB}(>D)$ is the MAB cumulative size distribution, we assume that:

$$N(>D) = k N_{MB}(>D), \qquad (6)$$

for $k < 1$. As $N_{MB}(>D)$, we used the de-biased size distributions computed in previous numerical studies[15]. They propose different possible distributions compatible with MAB observables, for the size range below one kilometer. We used the two extreme cases (n. 1 and n. 8 in[15], corresponding to different values of the impact threshold specific dispersion energy in their collisional model) as the lower and upper limits for $N_{MB}(>D)$. We call them $N_{B20.1}(>D)$ and $N_{B20.8}(>D)$, respectively.

The $k$ parameter is estimated by comparing the (biased) observed size distribution $N'(>D)$ of DCOs with $N_{MB}(>D)$ in the range between 1 and 10 km. The observed distribution $N'(>D)$ is built from the observed

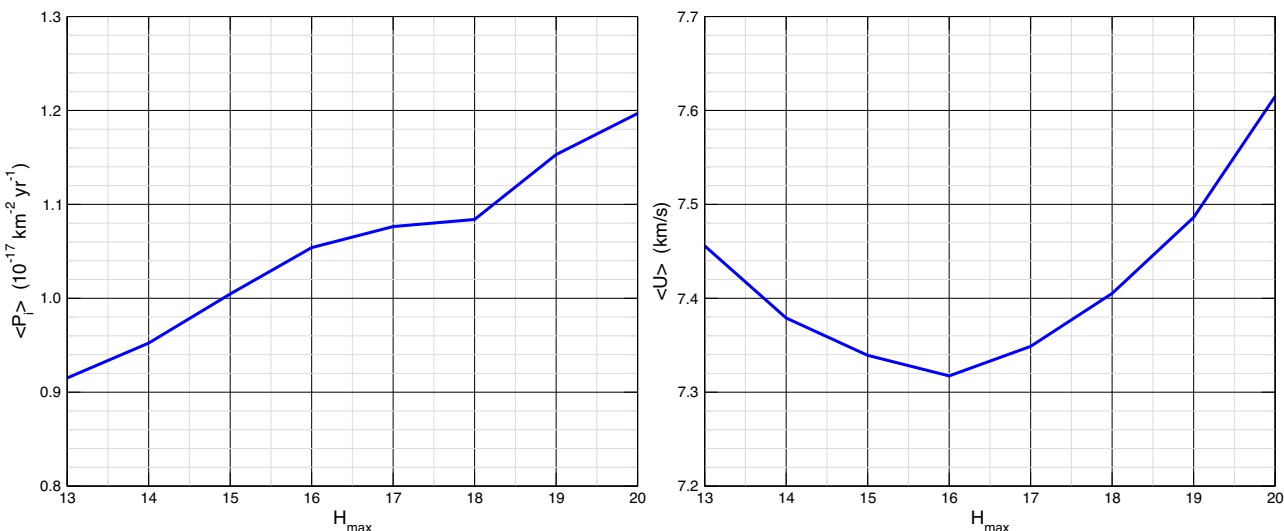

**Fig. 5 | Mean intrinsic probability of impact (left, in terms of $10^{-17}$ km$^{-2}$ yr$^{-1}$) and mean impact speed (right) between Didymos/Dimorphos system and DCO, versus the DCO maximum absolute magnitude.** Source Data are provided as a Source Data file.

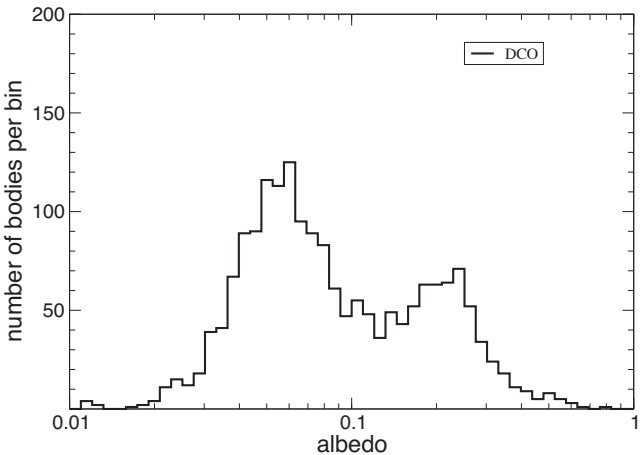

**Fig. 6 | Distribution of the albedo of DCOs according to AKARI database.** Source Data are provided as a Source Data file.

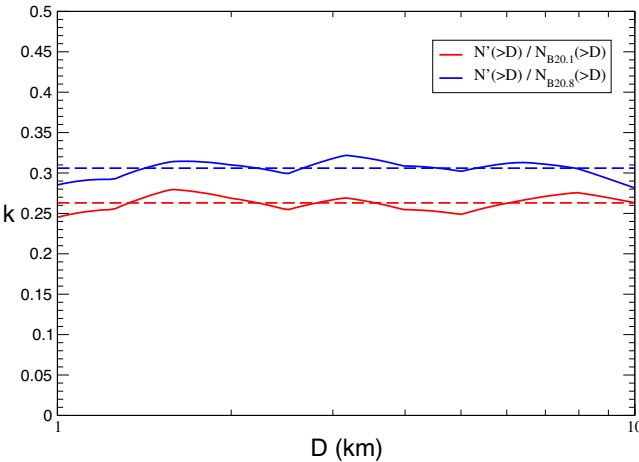

**Fig. 7 | Ratio between the biased distribution $N'(>D)$ and[15] de-biased distribution of MAB n. 1 (red) and n. 8 (blue), the interval between 1 km and 10 km.** The long-dashed lines are the mean values. Source Data are provided as a Source Data file.

distribution $N(<H)$ of the DCO absolute magnitudes, and the distribution of their albedos. The albedo data have been obtained by AKARI satellite database[33], selecting only the values of the DCOs. Figure 6 shows the distribution of DCO albedos obtained from AKARI survey. The typical bimodality, already known from IRAS data, is due to the contribution of C and S taxonomy classes.

Moreover, we assume that there is no correlation between absolute magnitudes and albedos. Such hypothesis may not be completely correct in general, but it is reasonably acceptable for the purpose of the present work. In this way, the number $dN'(>D)$ of the observed DCOs with diameter larger than $D$ and albedo in the interval $[\alpha, \alpha + d\alpha]$ is equal to:

$$dN'(>D) = N(<H(D, \alpha))\phi(\alpha)d\alpha \tag{7}$$

where $H(D, \alpha)$ is the absolute magnitude corresponding to the diameter $D$ and albedo $\alpha$:

$$H(D, \alpha) = 5\left(\log_{10} 1329 - \frac{1}{2}\log_{10}\alpha - \log_{10}D\right) \tag{8}$$

and $\phi(\alpha)$ is the (normalized) distribution of the albedo. In conclusion, the total number of observed DCOs with diameter larger than $D$, compatible with the observed distributions of absolute magnitudes and albedos, is:

$$N'(>D) = \int_0^1 N(<H(D, \alpha))\phi(\alpha)d\alpha \tag{9}$$

In practice, we evaluated integral (9) as:

$$N'(>D) \sim \frac{1}{n}\sum_k N(<H(D, \alpha_k)) \tag{10}$$

where $\alpha_k$, for $k = 1 \ldots n$, is the list of DCO albedos in our database, and $n$ is their total number.

Figure 7 shows the ratio between $N'(>D)$ and the distributions $N_{MB}(>D)$ in the range form 1 km to 10 km. The ratios are rather stable, with mean values $0.26 \pm 0.01$ and $0.31 \pm 0.01$ for the two cases n. 1 and n. 8 respectively. In conclusion, our extreme models of the de-biased distribution of the DCOs are:

$$\begin{aligned} N_{max}(>D) &= 0.26 N_{B20.1}(>D) \\ N_{min}(>D) &= 0.31 N_{B20.8}(>D) \end{aligned} \tag{11}$$

In Fig. 1 the different size distributions are plotted in the range between 20 cm and 100 km, along with the observed DCO distribution $N'(>D)$ divided by the two values of the scaling factor $k$. It is worthwhile pointing out that $N_{B20.1} > N_{B20.8}$ between 0.001 km and 10 km, within the interval of diameters of interest for the purposes of the present work.

### The orbit of Didymos and the Main Asteroid Belt

The asteroid belt is considered to have its inner limit at heliocentric distance $r_{MAB} = 2.06$ au. The orbit of the Didymos system has semi-major axis $a = 1.6425$ au, eccentricity $e = 0.3833$, and a low inclination with respect to the plane of the ecliptic, $i = 3.414$ deg. This configuration implies that part of the orbit is inside the inner Main Asteroid Belt. It is straightforward to calculate the fraction of time that Didymos spends in that region along its orbit. To do so, it is necessary to apply the classical relationship between the true anomaly ($v$) and the eccentric anomaly ($u$) of its orbit:

$$\tan\frac{v}{2} = \left(\frac{1+e}{1-e}\right)^{1/2}\tan\frac{u}{2}, \tag{12}$$

and the Kepler's equation, relating the body mean anomaly ($n$) to its eccentric anomaly:

$$n(t - t_0) = u - e\sin u, \tag{13}$$

where $t$ is a given time, and $t_0$ is the time of passage through the pericentre.

The values of $v$ corresponding to the enter and exit points from the inner belt may be derived through the JPL Horizons ephemeris generator. In that way, we find that Didymos spends a fraction of its orbital period beyond $r_{MAB}$ equal to 0.3556.

### Data availability

Source data are provided in the Source Data file. Source data are provided with this paper.

### Code availability

The codes used to compute collisional frequencies and impact velocity distributions implement the methods fully described in refs. 30,31. They are embedded in a multipurpose Java library, and researchers interested in using them are kindly invited to contact Dr. Aldo Dell'Oro.

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

## Acknowledgements

Project (PGC 2021) PID2021-125883NB-C21, by MICINN (Spanish Government): A.C.B., L.M.P., P.G.B. Call 2023 of the Italian National Institute for Astrophysics (INAF, act n. 38/2023): A.D.O. ESA funding through the Science Faculty—Funding reference ESA-SCI-SC-LE-191: P.G.B. "Margarita Salas" postdoctoral grant by the Spanish Ministry of University—NextGenerationEU: L.M.P. CIAPOS/2022/066 postdoctoral grant (European Social Fund. Generalitat Valenciana): L.M.P. Italian Space Agency (ASI) funding within the LICIACube project (ASI-INAF agreement n. 2019-31-HH.0): A.L., M.P. HERA project (ASI-INAF agreement n. 2022-8-HH.0): A.L., M.P.

## Author contributions

A.C.B. promoted the study and wrote most of the main manuscript, A.D.O. ran probability calculations and wrote most of the "Methods" section. A.C.B. and A.D.O. equally contributed through data analysis, manuscript set up and revision process. L.M.P. and P.G.B. equally contributed to the discussion of key issues, to the elaboration of graphics and writing suggestions. F.M. contributed with comments during the revision of the manuscript. P.M. and M.P. contributed with several suggestions during the writing process, and, together with S.J., A.L., J.B.V. contributed to the discussion that led to the manuscript itself.

## Competing interests

The authors declare no competing interests.
