## [Peer Review File · Nature Communications]

Recent collisional history of (65803) DidymosREVIEWER COMMENTS

Reviewer #1 (Remarks to the Author):

The manuscript presents an analysis of the recent collisional history of asteroid Didymos with the conclusion that collisions were responsible for surface features observed on Didymos and were more important than possible YORP-driven surface changes. The work builds upon the images of Didymos obtained by the DART and LICIACube spacecraft that revealed a non-spheroidal shape and heterogeneous surface with indications of material movement.

The methods are clearly described. The core of the work is estimating the collisional frequency with bodies that could alter the surface of Didymos to the observed state. However, the interpretation is based on indirect arguments: only the frequency of impacts is estimated, but their real effect on the surface is unknown. The impact frequency calculation is straightforward; it might be affected by uncertainties of the size distribution of small impactors, but the conclusion that small impacts are frequent seems to be robust. However, the evolution of the surface of Didymos is a complex process that is likely dominated by collisions but is also affected by YORP, although perhaps only indirectly. It is not clear if the conclusion that collisions are more important than YORP in surface processes is because Didymos is somehow exceptional or if we can generalize this conclusion to other similar near-Earth asteroids. It seems that this conclusion could be obtained even without the images from the DART mission - the collisional history estimates are not dependent on those. The discussion of why/if Didymos is exceptional is missing. For example, was the shape of Ryugu and Bennu also affected by collisions in the same way as Didymos? If yes, why are the shapes different? If not, what makes Didymos exceptional?

I strongly recommend discussing the results obtained for Didymos in the context of similar near-Earth asteroids and the role of YORP. Even if YORP is not directly responsible for the observed surface morphology of Didymos, it was needed to spin up Didymos and create Dimorphos, right? And what is the role of fast rotation? Is it necessary to create observed features by collisions, or would the effect of collisions be the same on a slowly rotating body?

Minor comments:

p. 2: "Initial estimations of the body equatorial extents [3] implied a non-equilibrium state [4]." - What is a non-equilibrium state?

p. 2: "...parameter space compatible with both equilibrium and equatorial instability." - So it is unclear if the surface is or is not stable?

p.2: "...indicating both spin up and down..." - only acceleration of the rotation has been observed so far, no spin down yet

p. 3: "... angular acceleration of $\alpha = d\omega/dt = (0.42 - 6.3) \times 10^{-6} \text{ deg/day}^2$ [16]." - The value of α in that paper is negative, so the rotation is decelerating.

p. 4: "...may be assumed to be $(0.5 - 1.0) \times 10^{-6} \text{ deg/day}^2$..." - this value is not larger than the range for Ryugu.

Fig. 1: The distribution in [13] does not go below $D = 10$ m. Here the distribution goes well below 1 m and the B20.1 and B20.8 lines seem to converge. Is the extrapolation to small sizes realistic? What about the uncertainty? How does it affect your results?

Fig. 3: The PDF does not show the right panel.

p. 11, eq. (11): I guess there should be " $N_B(>D)$ " instead of " $N_B(<D)$ "

Fig. 5: The PDF does not show the right panel.

Reviewer #2 (Remarks to the Author):

This is a very nice piece of work, extremely well written and easy to read. It explores an alternative explanation for the shape and surface features observed in Didymos, the primary of the binary system recently visited (and impacted) by the DART mission, and target of the ESA Hera mission. The authors conclude that collisions dominate over global deformation due to YORP, due to the fact that the asteroid spends about 1/3 of its orbit within the inner asteroid belt.

Although there are a number of assumptions that are needed to conduct the research (due to the evident unknowns), these are reasonable and well justified. The Methods section provides all the details in order to fully reproduce author's computations.

The paper presents original research and novelty, and therefore I recommend it for publication only after very minor revision.

My comments/corrections provided below:

- Page 2: about the reference to the DRA, it is only available for the DART team, you need username and password. Is this ok for a reference? Check Nature policy about this.
- Page 2, end of third paragraph: ...would not be surprising that they are also responsible for ...
- Page 3, first line of 2.1: .. torque based on asymmetric solar..
- Page 3, second paragraph of 2.1: ... surface area normal to the direction...
- Page 6, Figure 2: please provide the numbers and the tick marks for the y-axis.
- Page 7, Figure 3 (caption): Number '1' and '8' refer to the extreme...
- Page 9, last paragraph: ... described in previous work...
- Page 10: I am curious, why the notation $N_{B20.1}$ ($>D$)? It can be simply N_{B1} and N_{B8} . Why the 20?
- References: some references have the DOI, while others don't. I recommend unifying criteria.

Reviewer #3 (Remarks to the Author):

Review of the Nature Communications manuscript NCOMMS-23-55485 entitled "The (recent) collisional history of (65803) Didymos".

In this manuscript, the authors investigate the collisional environment of this asteroid and estimate the probability of collision with multi-size potential impactors. They analyze the possibility that such impacts produced the surface features observed on Didymos by comparing collisional intervals with estimated times for surface destabilization by the YORP effect.

The results are very interesting and significant; the text is clear and well-written, and the methods and procedures used throughout the manuscript are valid. However, it seems to me that the authors need to explain some points and assumptions in more detail to accept the manuscript for publication.

The first point needing clarification is the value of the mean intrinsic collision probability obtained for Didymos. The authors report a value that is 3-4 times larger than the mean value for the main asteroid belt (for example, see Table 1 in Davis et al. 2002, in comparison with your value of $0.9-1.2 \times 10^{-17} \text{ km}^{-2} \text{ yr}^{-1}$). This is strange because Didymos has an eccentric orbit and crosses the Near-Earth Asteroids region, the Hungaria asteroid region, and the inner belt, where it spends, in

the last zone, the most populated of the three, only one-third of its orbital period.

Another important point to clarify is in section 4, Methods, where you state that your calculation is based on three dynamical hypotheses, two of them being that the arguments of pericenter and longitudes of nodes change uniformly with time and that the angles of different orbits are not cross-correlated. However, the algorithm you used in your calculations (Dell'Oro & Paolicchi 1998; Dell'Oro 2017) can handle those particularities without any problem. In fact, the method of Dell'Oro & Paolicchi was used to study the collisional process of bodies in resonant orbits like Trojans and Plutinos (for example, Dell'Oro et al. 1998). Could you clarify this point in the text?

On the other hand, you find two values for different H_{max} , but any method used to calculate only uses the orbital elements of the population of projectiles. So, I must assume that you are using a projectile population that does not reach a level of completeness good enough to assure a good sampling of the orbital elements, and as a result, different values of are obtained depending on how the projectile population is chosen. Perhaps it is better to choose a good sample of the population where it is possible to assure completeness (large objects?) and use their orbital elements as representative of the element distribution of the population to make the calculations. Have you considered obtaining using a direct numerical integration method like, for example, Marzari et al. (1996)?

Finally, you use a cumulative size distribution for the projectile population, which is a fraction of the main belt's, but Didymos can interact only with objects in the inner belt, which is a zone strongly affected by resonances and has a size distribution that does not resemble that of the main belt. Have you considered any other way to estimate the population in the inner belt other than taking a simple fraction of the main belt?

Answers to reviewers.

Reviewer #1 (Remarks to the Author):

The manuscript presents an analysis of the recent collisional history of asteroid Didymos with the conclusion that collisions were responsible for surface features observed on Didymos and were more important than possible YORP-driven surface changes. The work builds upon the images of Didymos obtained by the DART and LICIACube spacecraft that revealed a non-spheroidal shape and heterogeneous surface with indications of material movement.

The methods are clearly described. The core of the work is estimating the collisional frequency with bodies that could alter the surface of Didymos to the observed state. However, the interpretation is based on indirect arguments: only the frequency of impacts is estimated, but their real effect on the surface is unknown. The impact frequency calculation is straightforward; it might be affected by uncertainties of the size distribution of small impactors, but the conclusion that small impacts are frequent seems to be robust. However, the evolution of the surface of Didymos is a complex process that is likely dominated by collisions but is also affected by YORP, although perhaps only indirectly. It is not clear if the conclusion that collisions are more important than YORP in surface processes is because Didymos is somehow exceptional or if we can generalize this conclusion to other similar near-Earth asteroids. It seems that this conclusion could be obtained even without the images from the DART mission - the collisional history estimates are not dependent on those. The discussion of why/if Didymos is exceptional is missing. For example, was the shape of Ryugu and Bennu also affected by collisions in the same way as Didymos? If yes, why are the shapes different? If not, what makes Didymos exceptional? I strongly recommend discussing the results obtained for Didymos in the context of similar near-Earth asteroids and the role of YORP. Even if YORP is not directly responsible for the observed surface morphology of Didymos, it was needed to spin up Didymos and create Dimorphos, right? And what is the role of fast rotation? Is it necessary to create observed features by collisions, or would the effect of collisions be the same on a slowly rotating body?

A: We thank this reviewer for their comments and for triggering the discussion on this topic. In principle, it is a bit beyond the scope of this paper to enter the discussion on the formation of the Didymos system. We think that we have more questions than answers on this topic. However, we now added some comments on that at the end of the main manuscript, leaving conclusions to the reader.

Minor comments:

p. 2: "Initial estimations of the body equatorial extents [3] implied a non-equilibrium state [4]." - What is a non-equilibrium state?

A: By non-equilibrium state we mean a rotational state which is no longer compatible with a surface of equilibrium at the equator, so mass can be lifted off.

p. 2: "...parameter space compatible with both equilibrium and equatorial instability." - So it is unclear if the surface is or is not stable?

A: This is correct. With the current available estimation of the equatorial extents of Didymos and its mass, instability is a possible state at the equatorial region.

p.2: "...indicating both spin up and down..." - only acceleration of the rotation has been observed so far, no spin down yet.

A: It is true that direct measurements are only reporting positive acceleration. We thank the reviewer for remarking this. Nevertheless, asteroid Ryugu has been estimated to be decelerating, as pointed out by the reviewer in their next comment. We modified the text accordingly and added a short comment on that.

p. 3: "... angular acceleration of $\alpha = d\omega/dt = (0.42 - 6.3) \times 10^{-6} \text{ deg/day}^2$ [16]." - The value of α in that paper is negative, so the rotation is decelerating.

A: The sign of the acceleration is now correctly reported in the text. We are grateful to the reviewer for pointing that out.

p. 4: "...may be assumed to be $(0.5 - 1.0) \times 10^{-6} \text{ deg/day}^2$..." - this value is not larger than the range for Ryugu.

A: We thank the reviewer once again for spotting this typo, we mean $(0.5 - 10) \times 10^{-6} \text{ deg/day}^2$.

Fig. 1: The distribution in [13] does not go below $D = 10 \text{ m}$. Here the distribution goes well below 1 m and the B20.1 and B20.8 lines seem to converge. Is the extrapolation to small sizes realistic? What about the uncertainty? How does it affect your results?

A: The extrapolation to small sizes (below 10 m) is based on the assumption that the size frequency distribution (SFD) is in a collisional steady state. We have no indication that the distribution would change below 10 m . Nevertheless, the reviewer is right that the real distribution below 1 km is unknown due to observational incompleteness. Though, we can't see any other way to overcome that than assuming that -below that size- collisional evolution did lead to the expected power-law distribution, as described by Bottke et al. (2020) (as the last of many possible references).

The convergence of the two MBA size distributions is an intrinsic characteristic of the model for the de-biased population by Bottke et al. (2020). This is due to the circumstance that the two distributions derive from two different Q_D^* functions in the strength regime, with different slopes. The uncertainty in our scaling factor k , computed comparing the DCO and the Bottke et al. (2020) distributions in the range $1\text{-}10 \text{ km}$, can be estimated of the order of 10% . Therefore the major source of uncertainty comes from the difference between the two extreme cases 1 and 8 that cannot be resolved on the basis of the observational constraints. Such a difference results to be larger than a factor of two for $D=100 \text{ m}$. This uncertainty appears in Fig. 3 as the difference between the solid and long-dashed lines.

Fig. 3: The PDF does not show the right panel.

A: Fixed.

p. 11, eq. (11): I guess there should be " $N_B(>D)$ " instead of " $N_B(<D)$ "

A: We are grateful for spotting this typo.

Fig. 5: The PDF does not show the right panel.

A: Fixed.

Reviewer #2 (Remarks to the Author):

This is a very nice piece of work, extremely well written and easy to read. It explores an alternative explanation for the shape and surface features observed in Didymos, the primary of the binary system recently visited (and impacted) by the DART mission, and target of the ESA Hera mission. The authors conclude that collisions dominate over global deformation due to YORP, due to the fact that the asteroid spends about 1/3 of its orbit within the inner asteroid belt. Although there are a number of assumptions that are needed to conduct the research (due to the evident unknowns), these are reasonable and well justified. The Methods section provides all the details in order to fully reproduce author's computations.

The paper presents original research and novelty, and therefore I recommend it for publication only after very minor revision.

A: We thank this reviewer for their comments and thorough reading that allowed us to improve the text.

My comments/corrections provided below:

-Page 2: about the reference to the DRA, it is only available for the DART team, you need username and password. Is this ok for a reference? Check Nature policy about this.

A: In fact, this information was available internally in the DART team. We replaced the DRA reference with a paper by Naidu and Chesley submitted to PSJ and hopefully accepted before the definitive version of this manuscript. Unfortunately, no other public reference is available so far.

-Page 2, end of third paragraph: ...would not be surprising that they are also responsible for ...

-Page 3, first line of 2.1: .. torque based on asymmetric solar...

-Page 3, second paragraph of 2.1: ... surface area normal to the direction...

-Page 6, Figure 2: please provide the numbers and the tick marks for the y-axis.

-Page 7, Figure 3 (caption): Number '1' and '8' refer to the extreme...

-Page 9, last paragraph: ... described in previous work...

A: We thank the reviewer for pointing out those issues in the text, that we corrected accordingly.

-Page 10: I am curious, why the notation $N_{B20.1} (>D)$? It can be simply N_{B1} and N_{B8} . Why the 20?

A: B20 refers to the paper Bootke et al. (2020) AJ, 160, 14, that provides the MBA size distributions from which our DCO (Didymos Crossing Objects) size distributions have been derived.

- References: some references have the DOI, while others don't. I recommend unifying criteria.

A: Thanks again. We dropped the DOIs.

Reviewer #3 (Remarks to the Author):

Review of the Nature Communications manuscript NCOMMS-23-55485 entitled "The (recent) collisional history of (65803) Didymos".

In this manuscript, the authors investigate the collisional environment of this asteroid and estimate the probability of collision with multi-size potential impactors. They analyze the possibility that such impacts produced the surface features observed on Didymos by comparing collisional intervals with estimated times for surface destabilization by the YORP effect. The results are very interesting and significant; the text is clear and well-written, and the methods and procedures used throughout the manuscript are valid. However, it seems to me that the authors need to explain some points and assumptions in more detail to accept the manuscript for publication.

A: We thank this reviewer for their comments and for pointing out some items that we will further clarify both in the reply and -where needed- in the text itself.

The first point needing clarification is the value of the mean intrinsic collision probability obtained for Didymos. The authors report a value that is 3-4 times larger than the mean value for the main asteroid belt (for example, see Table 1 in Davis et al. 2002, in comparison with your value of $0.9-1.2e-17 \text{ km}^{-2} \text{ yr}^{-1}$). This is strange because Didymos has an eccentric orbit and crosses the Near-Earth Asteroids region, the Hungaria asteroid region, and the inner belt, where it spends, in the last zone, the most populated of the three, only one-third of its orbital period.

A: The P_{int} is already taking into account the fact that only 1/3 of the orbit is spent by Didymos in the inner belt. In fact, the intrinsic probability of collision P_i has been computed specifically for the Didymos-Dimorphos (DD) bodies and it takes into account the orbit of the binary asteroid and those of its potential projectiles. In this way, it automatically takes care of the fact that the system only spends part of its orbit within the belt where collisions can occur. In fact, P_i is computed from real orbits and the algorithm takes into account their partial or total overlapping. Moreover, the value of the mean intrinsic probability computed for the Didymos-Dimorphos system is only apparently high. The definition of mean intrinsic probability depends on the choice of the population of impactors over which the average is performed. Values reported in Davis et al. 2002, and in particular those computed by Farinella and Davis (1992), are the average of the intrinsic probabilities of collision for all possible pairs of main belt asteroids, regardless they have crossing or not crossing orbits. In this way, also couples of orbits with $P_i=0$ contribute to lowering the average (see also discussion in Bottke et al., 1994). We have computed the mean value of P_i taking into account only the orbits that can cross the orbit of Didymos (in other words, orbits for which P_i is non-zero), and in fact this value is used in combination with the distribution of DCOs (Didymos Crossing Objects) only. If we had calculated the value including all the Main Belt orbits, we would have obtained a lower value (about $3.6e-18$).

Another important point to clarify is in section 4, Methods, where you state that your calculation is based on three dynamical hypotheses, two of them being that the arguments of pericenter and longitudes of nodes change uniformly with time and that the angles of different orbits are not cross-correlated. However, the algorithm you used in your calculations (Dell'Oro & Paolicchi 1998; Dell'Oro 2017) can handle those particularities without any problem. In fact, the method of Dell'Oro & Paolicchi was used to study the collisional process of bodies in resonant orbits like Trojans and Plutinos (for example, Dell'Oro et al. 1998). Could you clarify this point in the text?

A: The dynamical hypotheses for the computation of the statistics of impact are the standard assumptions underlying the classical approach by Wetherill (1967).

There is no reason to believe that such assumptions are not adequate for the case at hand. In particular, the fact that Didymos (a NEA) and the large majority of impactors (MBAs) belong to two different dynamical regions of the Solar System entails that the relative orientations of the target-projectile orbits are random (fulfilling assumption 3). In other words, the use of a more detailed dynamical model is not justified and ultimately useless considering other uncertainties, especially that on the impactors size distribution. Even if more general, the methods of Dell'Oro & Paolicchi 1998, and Dell'Oro 2017 are, in this case, fully equivalent to Wetherill's approach.

On the other hand, you find two values for different H_{max} , but any method used to calculate only uses the orbital elements of the population of projectiles. So, I must assume that you are using a projectile population that does not reach a level of completeness good enough to assure a good sampling of the orbital elements, and as a result, different values of are obtained depending on how the projectile population is chosen. Perhaps it is better to choose a good sample of the population where it is possible to assure completeness (large objects?) and use their orbital elements as representative of the element distribution of the population to make the calculations. Have you considered obtaining using a direct numerical integration method like, for example, Marzari et al. (1996)?

A: We thank the reviewer for raising this point and giving us the possibility to point out the details of the calculation. We have now further clarified in the text the procedure followed to mitigate the possible effects of observational biases. In fact, the method is exactly the one suggested by the reviewer himself, that is to identify a group of orbits that can be considered complete and reasonably representative of the global orbital distribution of DCOs. In this case, numerical integration of orbits does not help to solve the problems related to observational biases in the orbital distribution.

Finally, you use a cumulative size distribution for the projectile population, which is a fraction of the main belt's, but Didymos can interact only with objects in the inner belt, which is a zone strongly affected by resonances and has a size distribution that does not resemble that of the main belt. Have you considered any other way to estimate the population in the inner belt other than taking a simple fraction of the main belt?

A: This is an interesting point. The only available information about the SFD of the Didymos system impactors is the one that can be derived by the distribution of the DCOs. Beyond the completeness size, extrapolation to smaller sizes can only be done by taking into account the corresponding collisional steady state population. Based on past research on collisional evolution, we think that -whatever the size distribution is at large sizes (which is the domain at which completeness is achieved in the inner belt: $D > \sim 1$ km), the slope of the SFD, once steady state is reached, is fixed by the collisional cascade itself and by the collisional response of bodies (as explained in O'Brien & Greenberg, 2005). That is a standard procedure that is widely adopted in impact probability calculations. The fact that the inner belt population is dominated by the Flora family (which is understood to be collisionally relaxed) strengthens this approach.

REVIEWERS' COMMENTS

Reviewer #1 (Remarks to the Author):

I do not have any objections or comments to the revised version of the manuscript, and I recommend it for publication.

Reviewer #2 (Remarks to the Author):

The authors have fully addressed this reviewer's requests, and so I recommend the paper for publication in its current, revised form.

Reviewer #3 (Remarks to the Author):

Review of the Nature Communications manuscript NCOMMS-23-55485 entitled "The (recent) collisional history of (65803) Didymos".

In this revised version, the authors have satisfactorily addressed my suggestions, so I consider that the work can be published in its current form.